# Interference Aware Resource Control for 6G-Enabled Expanded IoT Networks

**DOI:** 10.3390/s23125649

**Published:** 2023-06-16

**Authors:** Ashu Taneja, Nayef Alqahtani, Ali Alqahtani

**Affiliations:** 1Chitkara University Institute of Engineering and Technology, Chitkara University, Rajpura 140401, Punjab, India; 2Department of Electrical Engineering, College of Engineering, King Faisal University, Al-Hofuf, Al-Ahsa 31982, Saudi Arabia; nmalqahtani@kfu.edu.sa; 3Department of Networks and Communications Engineering, College of Computer Science and Information Systems, Najran University, Najran 61441, Saudi Arabia; asalqahtany@nu.edu.sa

**Keywords:** 6G, next-generation IoT, interference, scheduling, resource control

## Abstract

Emerging consumer devices rely on the next generation IoT for connected support to undergo the much-needed digital transformation. The main challenge for next-generation IoT is to fulfil the requirements of robust connectivity, uniform coverage and scalability to reap the benefits of automation, integration and personalization. Next generation mobile networks, including beyond 5G and 6G technology, play an important role in delivering intelligent coordination and functionality among the consumer nodes. This paper presents a 6G-enabled scalable cell-free IoT network that guarantees uniform quality-of-service (QoS) to the proliferating wireless nodes or consumer devices. By enabling the optimal association of nodes with the APs, it offers efficient resource management. A scheduling algorithm is proposed for the cell-free model such that the interference caused by the neighbouring nodes and neighbouring APs is minimised. The mathematical formulations are obtained to carry out the performance analysis with different precoding schemes. Further, the allocation of pilots for obtaining the association with minimum interference is managed using different pilot lengths. It is observed that the proposed algorithm offers an improvement of 18.9% in achieved spectral efficiency using partial regularized zero-forcing (PRZF) precoding scheme at pilot length $\tau_{p}=10$. In the end, the performance comparison with two other models incorporating random scheduling and no scheduling at all is carried out. As compared to random scheduling, the proposed scheduling shows improvement of 10.9% in obtained spectral efficiency by 95% of the user nodes.

## 1. Introduction

The Internet of things (IoT) has changed the way we interact with and use our devices. It has a significant impact on the consumer electronics. IoT offers connectivity, automation, integration, personalization to the proliferating consumer devices. The functionality and efficiency of consumer devices, communication nodes, sensor nodes, mobile terminals is highly dependent on the next generation of IoT, which needs to be robust, intelligent and secure. Next generation IoT offers connected support to the consumers for personalized experience. But with the growing number of IoT devices comes challenges. Network congestion, reduced network speed, security risks, interoperability issues, compatability issues, data management, and battery management are the major challenges [1]. The power consumption or the energy overhead of the battery-powered IoT devices or consumer nodes led to severe energy crisis in next generation IoT network. For this, energy-efficient network designs coupled with energy harvesting techniques are required to offer sustainable solutions for next-gen IoT [2]. The heterogeneous IoT networks need to perform seamlessly, for which dynamic network frameworks are needed to support interoperability, integration and coordination [3]. Owing to the increasing number of IoT nodes and communicating terminals, the storage and security of the huge data are a challenge. The vulnerability of IoT devices and data to various threats require robust security mechanisms [4]. To overcome these challenges and to equip the next generation of IoT devices with powerful computing capabilities, intelligent decision-making, and better data management and security, future wireless technologies play an important role [5]. The sixth generation (6G) technology with extended fifth generation (5G) capabilities, offers high data rates, reduced latency and ultra-high reliability [6,7]. Multiple-antenna technologies [8], intelligent reflecting surfaces (IRSs) [9], heterogeneous networks (HetNets) [10], millimetre wave (mmWave) technology [11] are the breakthrough technologies of beyond 5G and 6G. The channels in 6G wireless networks involve non-stationarity, uncertain wireless propagation and high-mobility which require efficient modelling and characterisation for improved performance evaluation [12]. The channel measurements for different 6G wireless scenarios are different for all frequency bands [13]. MmWave technology suffers from shorter wavelengths and large signalling overhead [14,15] while HetNets cannot result in increased data rates after a certain cell density. Cell-free massive multiple-input–multiple-output (MIMO) is a key 6G technology that provides connected support to the communicating user nodes. With the help of multiple access points (APs) large number of nodes are being served by removing the cell boundaries such that the cell-edge traffic congestion is eliminated [16]. By distributing the radio access among multiple APs, it enhances network coverage, supports network scalability and reduces latency [17]. Through the cooperative processing capabilities of APs, it enables decentralisation such that the fault tolerance and network resilience is improved [18]. The 6G enabled cell-free network supports proliferating connected devices in the network with the help of cooperating APs, and thus is scalable [19]. This is different from MIMO and massive MIMO, which provide coverage to the nodes within the cell boundaries serviced by a centrally placed multi-antenna base station (BS) [20,21]. These suffer from the drawbacks of inter-cell interference, cell-edge traffic congestion and call drops owing to handovers [22]. The main idea of cell-free massive MIMO is to deploy large number of APs in a given coverage area to provide network connectivity and support to the IoT nodes [23]. The APs cooperate among themselves in order to serve the communicating nodes such that the network performance is enhanced [24,25]. The APs are connected through cloud processors called central processing units (CPUs) [26]. There are different network operations of cell-free systems, namely distributed and centralized. The signal processing takes place at the CPU in centralized operation where the AP acts as remote relays and send the information to the CPUs for processing [27]. In distributed operation, all the signal processing takes place at the AP. In order to provide stringent quality-of-service (QoS) to all the user nodes, a cell-free approach is used. It allows the coordination of a large number of distributed APs in a given coverage area for serving the enormous wireless devices [28]. Cell-free systems with single antenna nodes and APs are evaluated for performance in [29] for spectral efficiency. It is observed that, by equipping multiple antennas at the APs in a cell-free framework, the spectral efficiency can be further increased [30,31,32]. The cell-free approach is used recently in an expanded IoT network to provide extended communication support to the wireless nodes [33]. It aims to propose a scalable IoT network powered by cell-free method such that the energy efficiency is maximized using suitable power control. The role of the cell free system in sustainable IoT networks is highlighted in [34], where the energy is harvested to reduce the total energy consumption. In cell-free IoT networks, the beamforming is used in [35] to maximise the achievable rate using simultaneous wireless information and power transfer (SWIPT). The literature gives the comparison of conventional massive MIMO system with cell-free massive MIMO system [36,37,38]. It is highlighted that cell-free approach is more suitable for offering massive connectivity and high energy efficiency. The use of distributed APs in cell-free systems against the collocated antennas in massive MIMO results in improved performance. Different propagation environments are considered to carry out the performance comparison; for example, [39] assesses the cell-free IoT network with the Rayleigh channel model in addition to [40,41]. The signal processing in cell-free networks and their performance evaluation are carried out in [42,43,44,45]. The transmit precoding is considered in [42,43], which also uses optimal allocation of training symbols to achieve cooperative clustering. The channel estimation in cell-free networks is highlighted in [44], which aims for better efficiency with reduced normalized mean square error (NMSE). The use of receive combiners in cell-free networks for data detection results in the maximum gain [45]. The role of beamforming for minimum system interference is explored in [46]. The optimization of power control algorithms in the uplink of cell-free IoT systems is considered in [47] with efficient scalability analysis [19]. To serve the massive IoT nodes, the main challenge in a cell-free network is to coordinate or manage the APs and nodes optimally. The cooperation among the APs for coherent transmission with the wireless nodes need to be planned in such a way that interference is reduced. The user access for resources and associated APs requires efficient coordination and optimization such that the network throughput is improved in massive access scenarios. The intelligent algorithms involved in the optimization problems are reviewed in [48] for various application domains. For a multi-UAVs assisted communication scenario, ref. [49] investigates a task reallocation mechanism using optimization algorithm.

In this paper, a cell-free IoT network is proposed to meet the massive access needs of the enormous wireless nodes. For efficient network performance, a scheduling algorithm is proposed such that the interference due to neighbouring nodes and APs is minimised. The scalable precoding schemes are used under different cell-free network operations for maximum achieved spectral efficiency. The proposed communication scenario with an interference-aware scheduling algorithm is compared with system network incorporating random scheduling.

### Contributions and Outcomes

With the increasing number of IoT nodes, connected devices and the huge network traffic, the demand for seamless network connectivity has also increased. Thus, the need is to look for extended network capabilities in the future 6G wireless networks. This paper proposes a dynamic framework to offer massive connectivity in a 6G-enabled IoT network. The novel contributions of the paper are

A cell-free IoT network is proposed that supports the enormous wireless nodes with uniform coverage and QoS.An interference-aware scheduling algorithm is proposed that offers optimal resource management by associating APs to the nodes optimally using designed pilot allocation.The mathematical formulations are obtained for the achieved spectral efficiency under different precoding schemes for different cell-free operations.The system is evaluated for performance in terms of spectral efficiency achieved for different number of communication nodes, pilot lengths and precoding methods.The proposed cell-free network with the proposed scheduling mechanism is compared with other system models, one incorporating random scheduling and other not incorporating any scheduling.

The rest of the paper is organised as follows. Section 2 defines the system model along with the mathematical formulations for pilot transmission, channel estimation and data transmission. The precoding schemes used for the proposed model are also explained in Section 2. Section 3 gives the scheduling mechanism adopted for the association of APs and nodes followed by the proposed algorithm. The results are presented in Section 4 with a detailed discussion. Section 5 concludes the paper, giving a direction for the future scope. Table 1 lists the summary of notations used throughout the paper.

## 2. System Model

In this section, an IoT-enabled communication scenario involving large number of user nodes is considered where their communication is assisted by large number of access points (APs). Suppose *K* be the number of user nodes and *L* be the number of APs in a given IoT network. The user nodes are assumed to be equipped with a single antenna while the APs have multiple *N* antennas. The communication framework is shown in Figure 1. It assumes a user-centric cell-free approach in which a set of APs serves a particular user node. Let $A_{k}$ define a subset of APs that serve a particular user node *k* such that $A_{k}\subset \left\{1,2,...L\right\}$ and for every *k* and *l*
(1)$$q_{kl}=\left\{\;\begin{array}{c} 1\;\;\;\;\;\;\;l\in A_{k}\\ 0\;\;\;\;\;l\notin A_{k}\;\;\;\;\;\;\;\; \end{array}\right\}$$

The communication model that supports the transmission between each AP and each node adopts the block fading channel where the channel coefficients $h_{kl}$ between user *k* and AP *l* remain constant for each coherence block consisting of $\tau_{c}$ transmission symbols. The channel characteristics are estimated using uplink pilot training in the pilot transmission phase. The fading associated with the channels in each block is spatially correlated Rayleigh fading denoted as
(2)$$h_{kl}∼N\left(0,R_{kl}\right)$$
$h_{kl}$ are the independent and identically distributed channel elements drawn from Gaussian distribution with mean 0 and variance $R_{kl}$. Here, $R_{kl}$ is the spatial correlation matrix, which is related with the large scale fading given by
(3)$$\zeta_{kl}=tr\left(\frac{R_{kl}}{N}\right)$$

The large scale fading includes path loss with exponent α and shadowing $F_{kl}$ denoted as [50]
(4)$$\zeta_{kl}\left[dB\right]=-30.5-36.7log_{10}\left(\frac{r_{kl}}{1m}\right)+F_{kl}$$
with $r_{kl}$ being the distance between AP *l* and user node *k*. The network operation is assumed in which the $\tau_{c}$ symbols of coherence block comprise of $\tau_{p}$ symbols for pilot training and $\tau_{c}-\tau_{p}$ for data transmission.

### 2.1. Pilot Transmission and Channel Estimation

In the proposed system scenario, a set of orthogonal pilot sequences of length $\tau_{p}$ is used to obtain the channel statistics. Since the number of user nodes *K* is more than the number of orthogonal pilot symbols $\tau_{p}$, that is, $\tau_{p}<K$, each user cannot be provided with a unique pilot. Thus, the pilots are shared among the user nodes. The nodes that share the same pilots are referred to as co-pilot nodes. Let us define $U_{t}$ representing a set of nodes allocated pilot *t*. Also, $t_{k}\in \left\{1,2,....\tau_{p}\right\}$ denotes the index of pilots allotted to node *k*. Each user node *k* transmits the pilot *t* with pilot transmit power $\eta_{p}$. The signal received in this phase by the *l*th AP is given by
(5)$$y_{tl}^{p}=\;\;\sum_{i=1}^{K}\sqrt{\tau_{p}\eta_{p}}\;h_{il}\;+\;n_{tl}$$
where, $n_{tl}$ is the receiver noise such that $n_{tl}∼N\left(0,\sigma^{2}I_{N}\right)$. The user nodes in the set $U_{t_{k}}$ transmit the pilot $t_{k}$ which is received by the *l*th AP
(6)$$y_{t_{k}l}^{p}=\;\;\sum_{i\in U_{t_{k}}}\sqrt{\tau_{p}\eta_{p}}\;h_{il}\;+\;n_{t_{k}l}$$

In order to obtain the channel estimates ${\hat{h}}_{kl}$, minimum mean square error (MMSE) estimation method is used [21]
(7)$${\hat{h}}_{kl}=\;\sqrt{\tau_{p}\eta_{p}}R_{kl}\varphi_{t_{k}l}^{-1}\;y_{t_{k}l}^{p}\;$$
where $R_{kl}$ is the spatial correlation matrix which is equal to $E\left\{h_{kl}h_{kl}^{H}\right\}$, $\varphi_{t_{k}l}=E\left\{y_{t_{k}l}^{p}{\left(y_{t_{k}l}^{p}\right)}^{H}\right\}=\sum_{i\in U_{t_{k}}}\tau_{p}\eta_{p}R_{il}+\sigma^{2}I_{N}$ is the correlation matrix of the received signal.

### 2.2. Data Transmission

For the data transmission in the downlink, AP *l* transmits the data to user node *k*, which is precoded using the precoder $w_{kl}$ given below:(8)$$w_{kl}=\frac{{\bar{w}}_{kl}}{\sqrt{E\left\{\left|\right|{\bar{w}}_{kl}{\left|\right|}^{2}\right\}}}$$

The precoder is selected such that $E\left\{\left|\right|w_{kl}{\left|\right|}^{2}\right\}=1$. The role of precoder is to cancel the interference caused by the neighbouring user nodes. The transmission to the intended user may be interference to the other users. The interference can be mitigated using optimal precoding schemes [51]. In the centralized operation, all the signal processing including precoding is performed at the CPU, where interference can be cancelled by varying the transmit powers and phases of the transmitting APs. In distributed operation, where the signal processing takes place at the AP, suitable precoding schemes are used [52]. Let $x_{i}$ be the transmitted data signal intended for user node *k*. The signal received by user node *k* from the transmission of AP *l* is
(9)$$y_{k}^{DL}=\sum_{l=1}^{L}h_{kl}^{H}\sum_{i=1}^{K}\eta_{il}w_{il}x_{i}+n_{k}$$
where $\eta_{il}$ is the power AP *l* assigned to user node *i* with $\eta_{DL}$ being the maximum transmission power of each AP in the downlink. Depending on the network operation, different transmit precoding methods or schemes can be used [39]. Two main schemes used in the distributed operation are maximal ratio (MR) precoding and local partial minimum mean square error (LPMMSE) precoding.

MR precoding—The scalable precoding scheme to cancel out the interference caused by the AP is MR precoding, which is described by the equation below:(10)$$w_{il}^{MR}=\sqrt{\eta_{i}}\frac{{\hat{h}}_{il}}{\sqrt{E\left\{\left|\right|{\hat{h}}_{il}{\left|\right|}^{2}\right\}}}$$

LPMMSE precoding—This precoding scheme is the optimal and scalable extension of local minimum mean square error (LMMSE) precoding where only the nodes serviced by AP are considered [53].
(11)$$w_{il}^{LPMMSE}=\eta_{i}{\left(\sum_{i\in U_{l}}\eta_{il}\left({\hat{h}}_{il}{{\hat{h}}_{il}}^{H}+C_{il}\right)+\sigma^{2}I_{N}\right)}^{-1}{\hat{h}}_{il}$$
where $C_{il}=E\left\{{\tilde{h}}_{il\;}{{\tilde{h}}_{il}}^{H}\right\}$ is the error correlation matrix with ${\tilde{h}}_{il\;}=h_{il}-{\hat{h}}_{il}$ and $U_{l}$ specifies the set of user nodes served by AP *l*.

In the centralized operation of cell-free systems, the scalable precoding scheme used is partial regularized zero forcing (PRZF) precoding.
(12)$$w_{il}^{PRZF}={\left[{\hat{H}}_{A_{k}}{\left({\hat{H}}_{A_{k}\;}^{H}\;{\hat{H}}_{A_{k}}+\sigma^{2}P_{A_{k}}^{-1}\right)}^{-1}\right]}_{:,1}$$
where ${\hat{H}}_{A_{k}}=\sum_{i}{\hat{h}}_{il}$ for $i\in A_{k}$ and $P_{A_{k}}=diag\left(\eta_{il}\right)$

Another scalable precoding for centralized cell-free operation is partial minimum mean square error (PMMSE) precoding.
(13)$$w_{il}^{PMMSE}=\eta_{i}{\left(\sum_{i\in A_{k}}\eta_{il}{\hat{h}}_{il}{{\hat{h}}_{il}}^{H}+\sum_{i\in A_{k}}\eta_{il}C_{il}+\sigma^{2}I_{N}\right)}^{-1}{\hat{h}}_{il}$$

The achievable sum spectral efficiency is given by
(14)$${SE}_{k}=\left(1-\frac{\tau_{p}}{\tau_{c}}\right){log}_{2}\left(1+\;{SINR}_{k}\right)$$
(15)$$SINR_{k}=\frac{{\left|E\left\{\sum_{l=1}^{L}\eta_{kl}h_{kl}^{H}w_{kl}\right\}\right|}^{2}}{\sum_{i=1}^{K}{\left|E\left\{\sum_{l=1}^{L}\eta_{il}h_{kl}^{H}w_{il}\right\}\right|}^{2}-{\left|E\left\{\sum_{l=1}^{L}\eta_{kl}h_{kl}^{H}w_{kl}\right\}\right|}^{2}}$$

## 3. Interference-Aware Scheduling of APs and Nodes

In the communication scenario proposed in the paper, a large number of IoT nodes, mobile users and network terminals communicate through a set of APs. Since there are a large number of user nodes in the considered setup, each node is served by multiple APs. Also, each AP serves a multitude of user nodes. The interference is caused by the pilot reuse in the communication scenario. The user nodes sharing the same pilot are known as co-pilot user nodes and often led to interference with the intended recipient, known as pilot contamination, which degrades the channel estimation quality.

Let us consider a user node *k* that is assigned a pilot τ where $\tau \in \left\{1,2....\tau_{p}\right\}$. Since a set of fixed length orthogonal pilots are used, the pilots are reused among the communicating nodes. A set of nodes that are allocated the pilot τ except node *k* are termed as interfering nodes to node *k* and are denoted as $U_{\tau ,k}$. Also, a node *k* with pilot τ has interfering APs apart from the intended AP *l*. The set of interfering APs of node *k* is denoted as $L_{\tau ,k}$.
(16)$$U_{\tau ,k}=U_{\tau }\setminus \left\{k\right\}$$
(17)$$L_{\tau ,k}=\underset{i\in U_{\tau ,k}}{⋃}A_{i}\setminus A_{k}$$

To determine the strength of the intended AP-node link considering the interference caused by the interfering nodes and the interfering APs, a metric is defined as [54]
(18)$$I_{\tau ,k,l}=\zeta_{kl}-\max_{i\in U_{\tau ,k}}\zeta_{il}.\max_{j\in L_{\tau ,k}}\zeta_{kj}$$
for a particular node *k*, AP *l* and pilot τ.

### Proposed Algorithm

The proposed algorithm allows the optimal scheduling of nodes and APs and their association based on minimum interference due to neighbouring APs and neighbouring nodes on the intended AP-node link. First of all, each node *k* selects the AP *l*, which has the strongest channel gain to it. To accommodate the increasing user density, orthogonal pilots of length $\tau_{p}$ are reused among the user nodes. To avoid the interference, each AP can serve a maximum of $\tau_{p}$ sensor nodes. But there is a fair possibility of each AP serving more than $\tau_{p}$ nodes. Let us define $L_{o}$, a set of APs that serve more than $\tau_{p}$ nodes. Next, for each AP in the set $L_{o}$, a set of cluster nodes are found. Let ${\bar{U}}_{l}$ define a set of user nodes, which is being served by AP $l\in L_{o}$. Next, for each node in ${\bar{U}}_{l}$, the alternate AP *l*${}^{\prime }$ is chosen based on minimum channel loss criteria $\Delta =\zeta_{kl}-\zeta_{kl^{\prime }}$. The user node *i*${}^{\prime }$ in ${\bar{U}}_{l}$ with the minimum channel loss is selected and associated with AP *l*. The association between the node and AP is enabled by setting $\rho_{i}$’l=1. This is repeated till the association is found for all APs in $L_{o}$. Next, the allocation of pilots to the nodes is carried out. Initially random pilot allocation is performed for all the nodes. Then, to avoid pilot contamination, a set $N_{k}$ containing the $\tau_{p}-1$ neighbouring nodes of node *k* is defined. For each $k\in N_{k}$, the interference metric is calculated using Equation (Equation 18) for all $t\in \left\{1,2....\tau_{p}\right\}$. The pilot *t*${}^{\prime }$ at which the interference metric is maximum for a given k−l link is allotted to that node *k*. This is repeated till all nodes in $N_{K}$ are assigned pilots. Thus, the process results in optimal node-link association with optimal pilot allocation. The steps of the proposed scheduling algorithm are given in Algorithm 1 and the flowchart is given in Figure 2.
**Algorithm 1:**Interference-aware scheduling algorithm
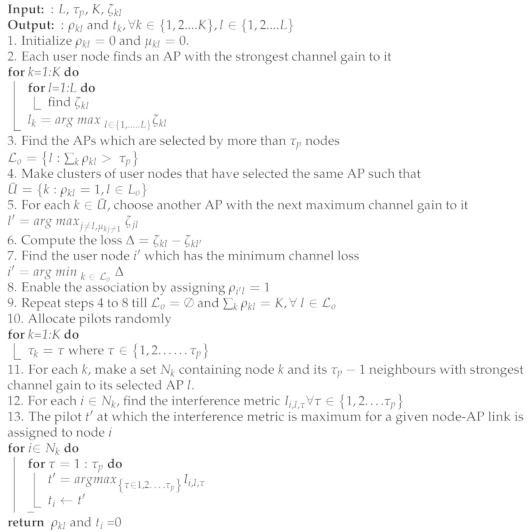


## 4. Results and Discussion

The proposed cell-free network with an expanded IoT framework is modelled in a MATLAB simulation environment [55]. The results of the performance evaluation are presented in this section. The simulation model considers deployment of large number of APs in a given area of 1 km × 1 km to give service to massive user nodes. The APs are installed with uniform linear array. The size of the array is N=4. The model considers a 3GPP microcell urban scenario with propagation parameters listed in Table 2. In the proposed model, an interference-aware scheduling scheme is presented whose performance evaluation is also carried out under different conditions. The performance parameters used for the evaluation of the proposed model include the number of connected nodes, number of APs, node locations, pilot length and spectral efficiency. The impact of different precoding schemes under different network operations on the performance is also investigated. A data or message flow diagram is presented in Figure 3.

The performance of the proposed cell-free IoT network is quantified in Figure 4 for spectral efficiency achieved by the IoT nodes. The four precoding schemes are used under different cell-free operations. LPMMSE and MR precoding are used for distributed operation while PRZF and PMMSE are used for centralized operation. The comparison is evaluated in Figure 4 in which the cumulative distribution function (CDF) of the spectral efficiency in the downlink operation is plotted. The spectral efficiency obtained is 2.949 bits/s/Hz for MR precoding and 8.592 bits/s/Hz for scalable LPMMSE precoding. For the centralized operation, the PRZF and PMMSE precoding have similar performance, which gives 9.207 bits/s/Hz downlink spectral efficiency.

In the proposed model, the scheduling of nodes and serving APs is achieved using the proposed scheme, which takes into account the interference caused by the neighbouring nodes as well as the interfering APs. The CDF is obtained for the downlink operation to assess the performance of the proposed algorithm in Figure 5 for the precoding schemes defined in Section 2. The incorporation of proposed scheduling method, which facilitates optimal association of sensor nodes and the APs results in improved system performance. It has been observed that centralized precoding outperforms the LPMMSE and MR precoding achieving 18.9% improvement in achieved spectral efficiency with the proposed scheduling.

The communication model with proposed scheduling is compared with two other system models, one that is not incorporating any user-AP association and another that is using random scheduling to assign the nodes with the APs. The performance of these three models is depicted in Figure 6, which plots the spectral efficiency achieved by 95% of the communication nodes. As compared to random scheduling, the proposed scheduling shows improvement of 10.9% in obtained spectral efficiency with PRZF precoding. The system model with no scheduling incorporated, the 95% likely achieved spectral efficiency is minimum, which is 0.5789 bits/s/Hz with MR precoding, 1.1105 bits/s/Hz with LPMMSE precoding and 1.6869 bits/s/Hz with PRZF and PMMSE precoding. The variation of average spectral efficiency with number of nodes is shown in Figure 7. As the number of nodes are increased from 20 to 100, the average spectral efficiency is decreased to 1.69 bits/s/Hz in MR, 3.9 bits/s/Hz in LPMMSE and 4.799 bits/s/Hz in the PRZF precoding scheme. The impact of pilot length $\tau_{p}$ on the average spectral efficiency of the system is obtained in Figure 8. The performance of different precoding schemes under different operations of the proposed cell-free model is evaluated with the scheduling schemes. It is observed that on increasing the pilot length, the spectral efficiency first increases and then it saturates. On increasing the pilot length, the interference is reduced resulting in improved spectral efficiency performance. The AP-node association is improved as more APs can serve each node. But a further increase in $\tau_{p}$ will lead to saturation as the transmission symbol length is reduced. Here also, the proposed scheduling method outperforms the random scheduling and no scheduling approaches being adopted in different system models, respectively. The maximum spectral efficiency of 8.712 bits/s/Hz is obtained with proposed scheduling algorithm and PRZF precoding scheme at $\tau_{p}=10$.

## 5. Conclusions

Next generation IoT offers connected support to the consumers for personalized experience. The digital transformation of emerging consumer devices rely on future wireless technologies. In this paper, a 6G-enabled scalable cell-free IoT network is proposed that aims for uniform coverage to the nodes in the network. An interference-aware scheduling algorithm is proposed that enables optimal association of nodes and APs with robust interference management. The spectral efficiency performance is obtained for different precoding schemes under different cell-free operations. It is observed that the obtained value is 2.949 bits/s/Hz for MR precoding and 8.592 bits/s/Hz for scalable LPMMSE precoding. For the centralized operation, the PRZF and PMMSE precoding gives 9.207 bits/s/Hz downlink spectral efficiency. The centralized precoding outperforms the LPMMSE and MR precoding achieving 18.9% improvement with proposed scheduling. Further, the comparison with other system models reveals that as compared to random scheduling, the proposed scheduling shows improvement of 10.9% in obtained spectral efficiency by 95% of the user nodes with PRZF precoding. Also, the 95% likely achieved spectral efficiency is the minimum for the system model with no scheduling incorporated.

The energy efficiency analysis of the proposed cell-free network model with interference-aware scheduling is left for future work. The energy management of the network is necessary to support green communication in future IoT networks.

## Figures and Tables

**Figure 1 sensors-23-05649-f001:**
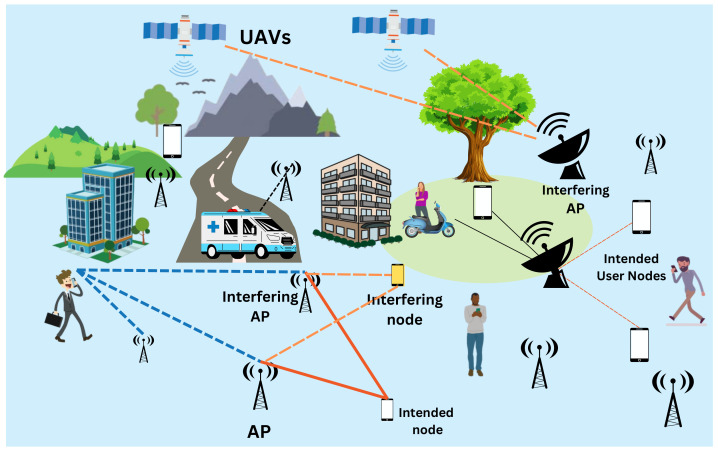
Proposed communication scenario showing intended AP-node transmission affected by interfering nodes and interfering APs.

**Figure 2 sensors-23-05649-f002:**
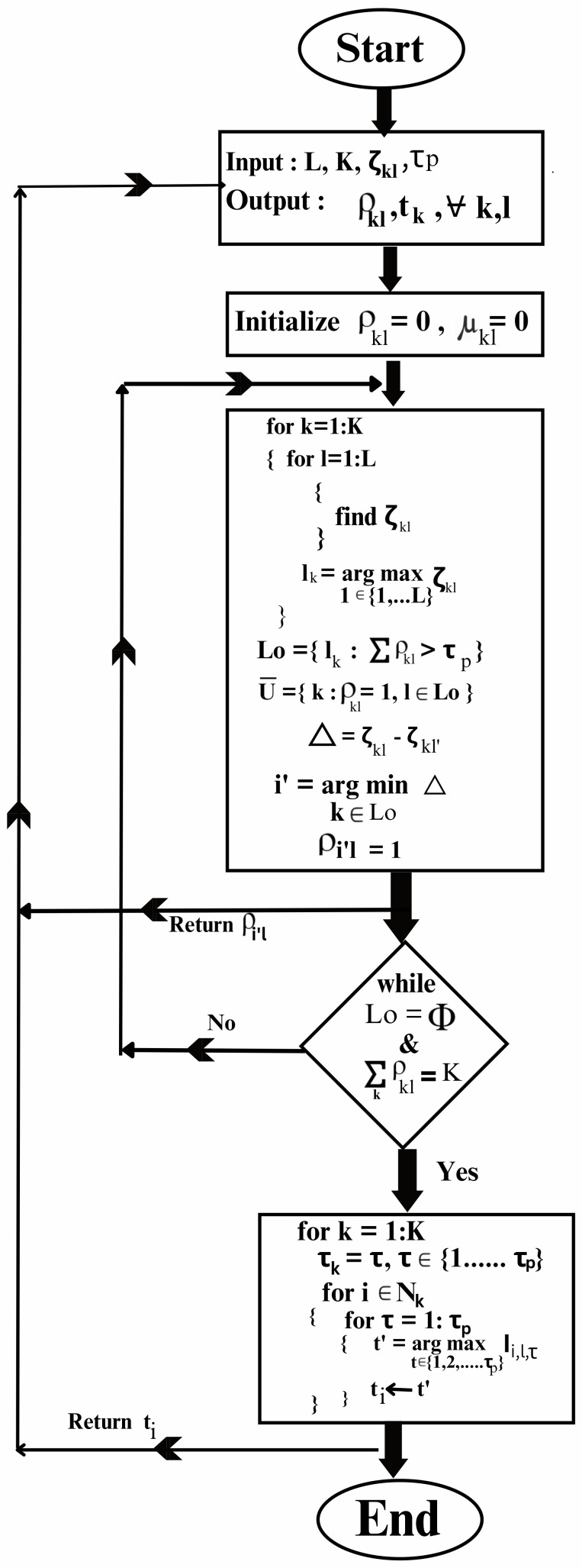
Flowchart of the proposed algorithm.

**Figure 3 sensors-23-05649-f003:**
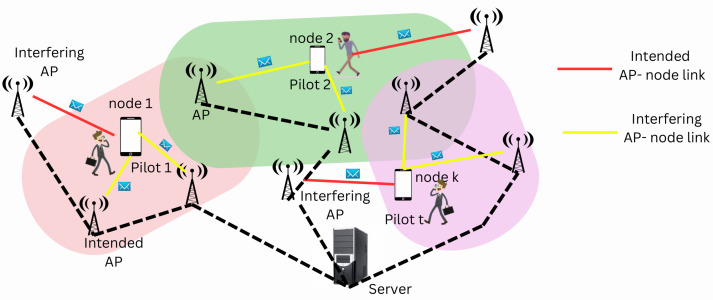
Illustration of data or message flow in the proposed network model.

**Figure 4 sensors-23-05649-f004:**
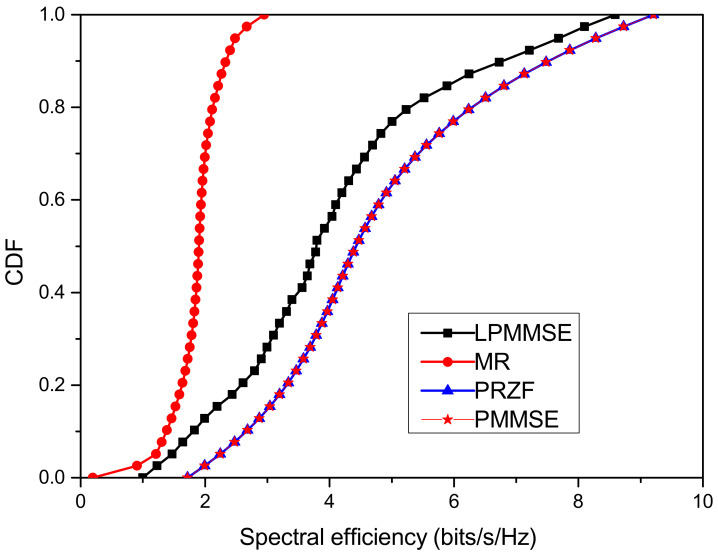
CDF of spectral efficiency achieved with precoding schemes under different operations.

**Figure 5 sensors-23-05649-f005:**
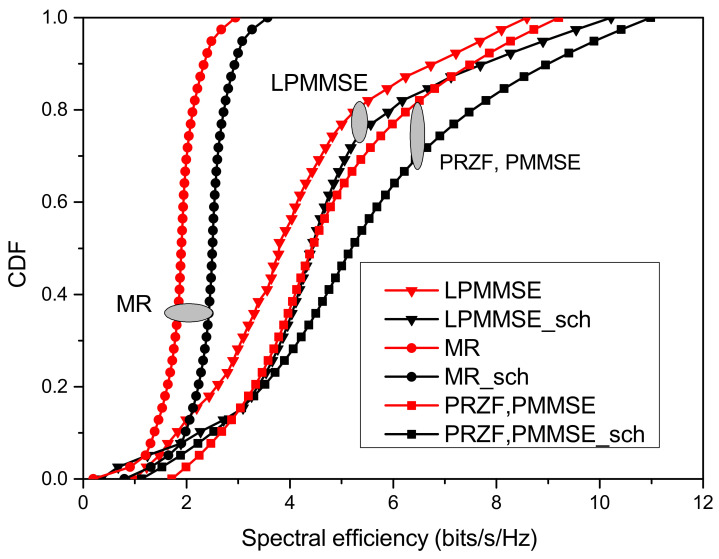
Performance of proposed scheduling scheme for different precoding methods.

**Figure 6 sensors-23-05649-f006:**
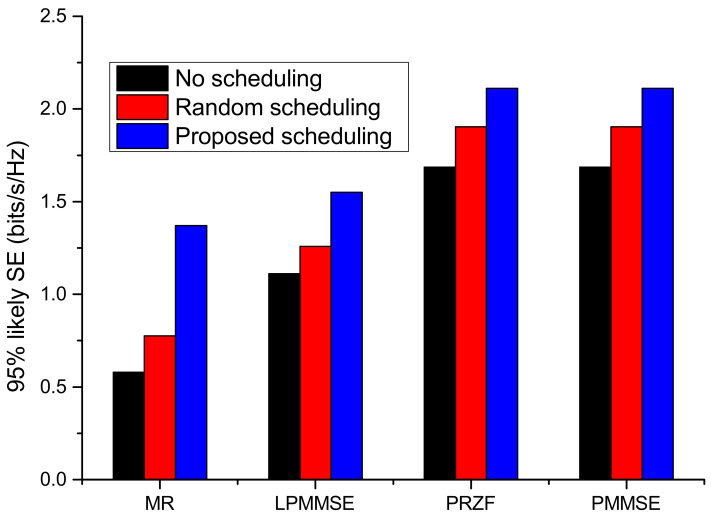
The 95% likely spectral efficiency for different scheduling schemes.

**Figure 7 sensors-23-05649-f007:**
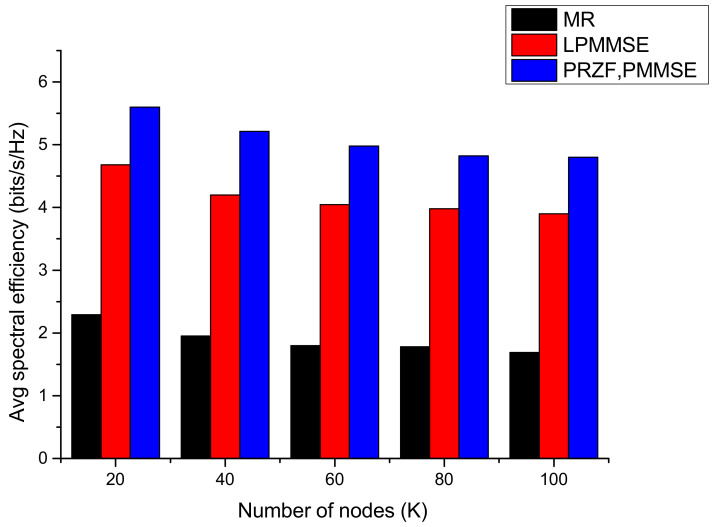
Average spectral efficiency as a function of number of nodes.

**Figure 8 sensors-23-05649-f008:**
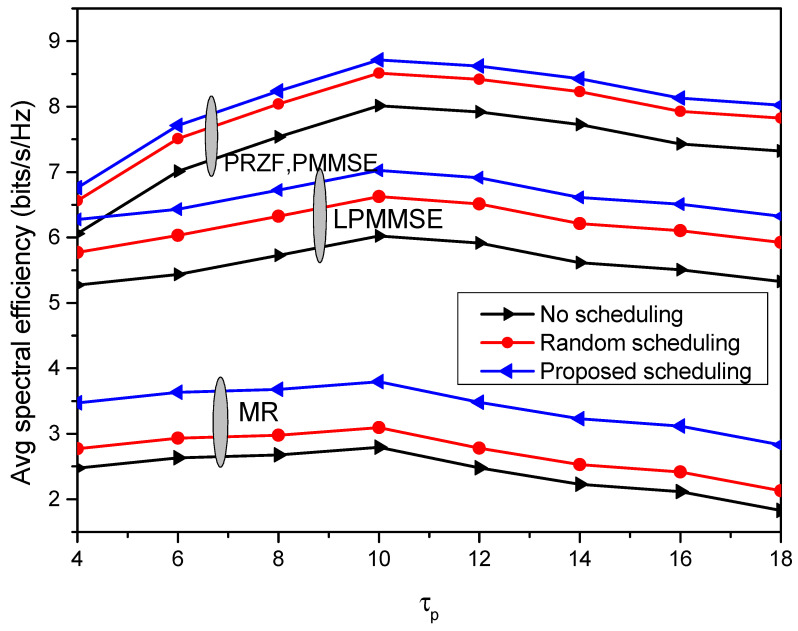
Comparison of different scheduling methods under different pilot lengths.

**Table 1 sensors-23-05649-t001:** Summary of notations.

Notation	Description
*L*	Number of APs
*N*	Number of antennas at each AP
*K*	Number of user nodes
α	Path loss exponent
$\eta_{k}$	Transmit power of user *k*
$\tau_{p}$	Pilot length
ψ	A set of $t_{p}$ pilot sequences
$x_{i}$	Transmitted signal
$A_{k}$	Set of APs serving node k
$n_{tl}$	Receiver noise
$N_{n}$	Receiver noise during pilot transmission phase
$h_{kl}$	Channel coefficients between node *k* and AP *l*
${\hat{h}}_{kl}$	Estimated channel coefficients
$\zeta_{kl}$	Large scale fading coefficients
$r_{kl}$	Distance between AP *l* and node *k*
$w_{kl}$	Precoding vector
$y_{k}^{DL}$	Received downlink signal
$U_{l}$	Set of nodes served by AP *l*
$R_{kl}$	Spatial correlation matrix
$\varphi_{t_{k}l}$	Correlation matrix of the received signal
$C_{il}$	Error correlation matrix
$L_{\tau ,k}$	Set of interfering APs of node *k*
$U_{\tau ,k}$	Interfering nodes to node *k*
$I_{\tau ,k,l}$	Interference metric

**Table 2 sensors-23-05649-t002:** Simulation parameters.

Parameters	Value	Parameters	Value
*N*	4	*L*	50
*K*	40	α	3.76 m
$\tau_{p}$	10	$\eta_{p}$	100 mW
$\tau_{c}$	200	$\eta_{k}$	100 mW
*B*	20 MHz	$\sigma^{2}$	−94 dBm
$B_{c}$	100 KHz	$T_{c}$	1 ms

## Data Availability

Not applicable.

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
