# Peer review of "Interference Aware Resource Control for 6G-Enabled Expanded IoT Networks"

_sensors, 2023, doi:10.3390/s23125649_

Round 1
Reviewer 1 Report
Authors propose a 6G enabled scalable cell-free IoT network that guarantees uniform quality-of-service (QoS) to the proliferating wireless nodes or consumer devices.
It is good research work of a very hot topic. But it needs some improvements to get accept the paper.
- Authors should cite some papers of 6G channel characteristics in order to provide better background to the reader such as:
Channel modeling and characteristics for 6G wireless communications, IEEE Network 35 (1), 296-303.2020
- Authors should include a paragraph at the end of the introduction section explaining the rest of the paper.
- Authors should include a message flow diagram in section 3 showing the communication between AP and nodes.
- In order to improve figure 2, authors should provide text in each text box, instead of adding formulas.
- Authors should provide a citation/reference to Matlab simulator.
- Authors should include their future work at the end of the conclusion section.
Reviewer 2 Report
The authors have presented a 6G enabled scalable cell-free IoT network to ensure uniform QoS for wireless nodes. They have worked on optimal node-AP association and scheduling algorithm to minimize interference, duly supported by mathematical formulations using different precoding schemes. The proposed algorithm improves spectral efficiency using PRZF precoding. Performance comparison with random and no scheduling models is also studied.
The authors may consider the following for further improvement in the manuscript:
- include a comprehensive overview of the research gaps in the field of next-generation IoT networks in the introduction and provide a more detailed discussion of the current state-of-the-art to highlight the novelty and significance of the proposed approach. Much work has already been carried out in this area.
- clearly define the technical requirements for robust connectivity, uniform coverage, and scalability in next-generation IoT networks. Include quantitative metrics and performance benchmarks for the evaluation of the effectiveness of the proposed work.
- Elaborate on how beyond 5G and 6G mobile networks specifically contribute to intelligent coordination and functionality among consumer nodes. Highlight such unique features and capabilities for the proposed cell-free IoT architecture.
- Add more clarity and technical depth to describe the 6G enabled scalable cell-free IoT network and provide a thorough discussion of the underlying principles, architectural components, and key design considerations needed to establish such a network. Address potential challenges and limitations.
- optimal node-AP association and resource management is inadequately explained.
- Provide a step-by-step derivation and explanation of the equations used and include assumptions made and provide justifications for the chosen precoding schemes and pilot lengths.
- The observed is not sufficiently supported. Conduct a rigorous to strengthen the validity of the results.
- The performance comparison with other models lacks and consider additional relevant metrics such as latency, energy efficiency, or network capacity to provide a more holistic assessment of the proposed algorithm.
- a thorough language review to identify the grammatical errors, revise sentence structures, and correct imprecise phrasing. Improve the overall readability and flow of the text.
a thorough language review to identify the grammatical errors, revise sentence structures, and correct imprecise phrasing. Improve the overall readability and flow of the text.
Reviewer 3 Report
Revise Opinion
The paper proposes a cell-free IoT network that supports 6G, which supports the enormous wireless nodes with uniform coverage and QoS. It also proposes an interference aware scheduling algorithm which can realize the optimal correlation between nodes and APs.
The article can be modified:
1. In Section 3.1, the sentence “The proposed scheduling algorithm along with the steps involved is given in Algorithm 1 and the flowchart is given in fig.” is incomplete. The content of figure 2 is not clear and the drawing is rough.
2. The scheduling algorithm proposed in this paper does not give an appropriate algorithm name to show the characteristics of this method.
3. In the experiment, it uses LPMMSE and MR precoding for distributed operation, and PRZF for centralized operation. It should add another precoding scheme for centralized operation in order to make a fairer comparison.
4. When discuss the optimal scheduling problem in Introduction, ‘A review on representative swarm intelligence algorithms for solving optimization problems’ and ‘Dynamic Reallocation Model of Multiple Unmanned Aerial Vehicle Tasks in Emergent Adjustment Scenarios’ would be useful.
5. The simulation model assumes that a large number of APs are installed in a uniform line array and the size of the array is 4. Different array sizes can be set to further demonstrate the advantages of the proposed scheduling scheme.
Some syntax error need to be modified.
Reviewer 4 Report
The author must justify equation (2) and clarify every symbol’s meaning.
Why are there blank spaces below Table 2?
The authors need to justify using the pilot length they considered in the study.
The abstract can be improved by adding other results in brief.
The symbol used for ‘kilo’ is not capitalized.
The figure at the sentence’s end should be as Fig. (not fig.)
The simulation parameters used in Table 2 should be justified.
The authors must follow a standard procedure to refer to Equations, Figures, and Tables.
In line 154, “the flowchart is given in fig” what does it mean?
Algorithm 1 needs to be written better. In the current version, the algorithm could be clearer to comprehend.
The flow chart in Fig. 2 needs revision to present it better. It is hard to follow the flow chart.
Round 2
Reviewer 1 Report
Authors have fixed all my comments. The paper is ready to be published.
Author Response
Detailed Response to Reviewer comments
Authors have fixed all my comments. The paper is ready to be published.
Response: Thank you so much.
Reviewer 2 Report
Authors have complied to the suggestions and the revised manuscript has been revised to attain satisfactory level of quality.
Author Response
Detailed Response to Reviewer comments
Authors have complied to the suggestions and the revised manuscript has been revised to attain satisfactory level of quality.
Response: Thank you so much.
Reviewer 4 Report
1. Please allow a space after a number in Table 2 (in 1ms).
2. Please allow a space after Fig. in line 196.
3. But I find KHz in Table 2.
4. The mathematical expression inside the flow chart needs to write better.
5. The new paragraph has not been started after equation (2). So, please do not allow indent.
6. The authors should include line numbers in the algorithm. In the current version, several statements are expanded to the second line. Line numbers can solve the problem. Also, the authors can find a better another approach to write the algorithm.
7. 1Km should be 1 km (in line 188)
8. The operations in the block diagram of flow chart 2 should improve further to understand better the functions used in the flow chart.
